# Antilymphoma Effect of Incomptine A: In Vivo, In Silico, and Toxicological Studies

**DOI:** 10.3390/molecules26216646

**Published:** 2021-11-02

**Authors:** Fernando Calzada, Elihú Bautista, Sergio Hidalgo-Figueroa, Normand García-Hernández, Elizabeth Barbosa, Claudia Velázquez, Rosa María Ordoñez-Razo, Angel Giovanni Arietta-García

**Affiliations:** 1Unidad de Investigación Médica en Farmacología, UMAE Hospital de Especialidades, 2º Piso, Centro Médico Nacional Siglo XXI, Instituto Mexicano del Seguro Social, Av. Cuauhtémoc 330, Col. Doctores, Mexico City CP 06725, Mexico; 2CONACYT—Consorcio de Investigación, Innovación y Desarrollo para las Zonas Áridas, Instituto Potosino de Investigación Científica y Tecnológica A. C., San Luis Potosí CP 78216, Mexico; francisco.bautista@ipicyt.edu.mx (E.B.); sergio.hidalgo@ipicyt.edu.mx (S.H.-F.); 3Unidad de Investigación Médica en Genética Humana, UMAE Hospital Pediatría, 2º Piso, Centro Médico Nacional Siglo XXI, Instituto Mexicano del Seguro Social, Av. Cuauhtémoc 330, Col. Doctores, Mexico City CP 06725, Mexico; normandgarcia@gmail.com (N.G.-H.); romaorr@yahoo.com.mx (R.M.O.-R.); angelo_arietta@hotmail.com (A.G.A.-G.); 4Sección de Estudios de Posgrado e Investigación, Escuela Superior de Medicina, Instituto Politécnico Nacional, Salvador Díaz Mirón esq. Plan de San Luis S/N, Miguel Hidalgo, Casco de Santo Tomas, Mexico City CP 11340, Mexico; rebc78@yahoo.com.mx; 5Área Académica de Farmacia, Instituto de Ciencias de la Salud, Universidad Autónoma del Estado de Hidalgo, Km 4.5, Carretera Pachuca-Tulancingo, Unidad Universitaria, Pachuca CP 42076, Mexico; cvg09@yahoo.com

**Keywords:** incomptine A, antilymphoma activity, molecular docking, acute toxicity, brine shrimp lethality, cancer, non-Hodgkin lymphoma

## Abstract

Incomptine A (**IA**) is a sesquiterpene lactone isolated from *Decachaeta incompta* that induces apoptosis, reactive oxygen species production, and a differential protein expression on the U-937 (diffuse histiocytic lymphoma) cell line. In this work, the antitumor potential of **IA** was investigated on Balb/c mice inoculated with U-937 cells and through the brine shrimp lethality (BSL) test. Furthermore, **IA** was subjected to molecular docking study using as targets proteins associated with processes of cancer as apoptosis, oxidative stress, and glycolytic metabolism. In addition to determining the potential toxicity of **IA** in human, its acute toxicity was performed in mice. Results reveals that **IA** showed high antilymphoma activity and BSL with an EC_50_ of 2.4 mg/kg and LC_50_ 16.7 µg/mL, respectively. The molecular docking study revealed that **IA** has strong interaction on all targets used. In the acute oral toxicity, **IA** had a LD_50_ of 149 mg/kg. The results showed that the activities of **IA** including antilymphoma activity, BSL, acute toxicity, and in silico interactions were close to the methotrexate, an anticancer drug used as positive control. These findings suggest that **IA** may serve as a candidate for the development of a new drug to combat lymphoma.

## 1. Introduction

Sesquiterpene lactones (SLs) are a group of phytochemicals present in species of the plant kingdom, specifically abundant in the family Asteraceae with more than 5000 different structures reported. They are a large group of low molecular weight specialized metabolites with a 15 carbon skeleton. SLs are considering as a class of natural compounds with pharmaceutical relevance that may be an alternative for cancer therapy [1,2]. SLs also have a wide range of biological activities such as antiamoebic, antigiardial, trypanocidal, antidiabetic, antibacterial, cytotoxic, antitumor, and phytotoxic [3,4,5,6,7,8].

In the case of antitumor activity, studies on their molecular mechanism action have reported that SLs exercise their antitumor properties through a change in the redox cell balance and their impact on various signaling pathways including the nuclear factor-kB (NF-kB) pathway and signal transducer and activator of transcription 3 (STAT3). These properties cause affectation in the cell cycle, increase apoptotic factors, decrease cellular invasion and metastasis factors, reduce the anti-apoptotic factors, and increase the production of free radicals associated with the induction of apoptosis [1,2]. In vivo antitumor activity of several SLs such as parthenin, arteminolides A–D, eupatoriopicrin, and parthenolide (**Pa**) has been reported [8]. Non-Hodgkin lymphomas (NHLs) are a heterogenous group of neoplasms or lymphoproliferative malignancies, derived from the basic cells of lymphoid tissue such as lymphocyte and histiocytes in any of their developmental stages. In 2015, nearly 4.3 million people were reported to have non-Hodgkin lymphoma (NHL), which accounted for 231,400 deaths around the world, predominantly in adults [9,10]. In the case of Mexico until 2016, NHL was the fourth most common cause of cancer [11]. Several drugs are available for its treatment including cyclophosphamide, prednisone, cisplatin, fludarabine, methotrexate (**MTX**), doxorubicin, vincristine, and etoposide. Unfortunately, in addition to the therapeutic properties, most of these drugs showed limited chemotherapeutic efficacy and have various side effects [12].

Previously, the fractionation from a CH_2_Cl_2_ extract of the leaves of *Decachaeta incompta* (DC.) R. M. King & H. Rob led to the isolation of four heliangolide-type sesquiterpene lactones, incomptines A–D among these incomptine A (**IA**) were the major antiprotozoal compound. In this sense, the antiprotozoal evaluation of six 8-acyl and four 8-alkyl incomptine A analogs showed that antiprotozoal properties against *Entamoeba histolytica and Giardia lamblia* trophozoites of these classes of compounds were related to the substituent at C-8 of the germacrene frame work [4,13,14]. Continuing with our search to obtain new antitumor agents, in the present work, incomptine A (**IA**) with α-methylene-γ-lactone that recently was reported as a specialized metabolite with cytotoxic activity [15], was utilized to evaluate its antilymphoma activity and acute toxicity on Balb/c mice. The brine shrimp lethality test was used as a bioindicator of cytotoxic activity, and molecular docking studies were used to uncover potential targets that explain the antilymphoma properties of **IA**.

## 2. Results

### 2.1. Antilymphoma, Lethality, and Toxic Activities of Incomptine A (**IA**)

The brine shrimp acute lethality test showed (Table 1) that the dichloromethane extract of *Decachaeta incompta* aerial parts (**DEDi**) was lethal to *Artemia salina* larvae with a LD_50_ value of 90.2 µg/mL, and its sesquiterpene lactone, incomptine A (**IA**, Figure 1), was five times more active than origin extract with a LD_50_ value of 16.7 µg/mL. In the case of methotrexate, an antilymphoma drug was used as the positive control and its LD_50_ value was 24.6 µg/mL. The results of antilymphoma activity in a mice model (Table 1) showed that all samples had significant antitumor properties on leukemic monocyte lymphoma (LML) with EC_50_ values of 75.0 mg/kg, 2.4 mg/kg, and 1.3 mg/kg for **DEDi**, incomptine A, and methotrexate, respectively. In the acute oral toxicity test, the **DEDi** had a LD_50_ of 1134.0 mg/kg, while the incomptine A had a LD_50_ of 149.0 mg/kg and the methotrexate LD_50_ was 335.0 mg/kg.

### 2.2. Molecular Docking Studies of Incomptine A (**IA**) 29 Several Selected Pharmacological Receptors Associated to Cancer

In a previous study, the cytotoxicity of **IA** was determined against the NHL cell line U-937 and the subsequent proteomic analysis showed a differential protein expression. Some of these proteins expressed differentially are involved in cellular processes crucial for the survival of cancer cells such as cytoskeleton structuration, glycolytic metabolism, apoptosis induction, and oxidative stress. Therefore, in the present work, in order to understand and how **IA** can affect these molecular targets, a docking study was carried out and compared against **MTX**, used as a drug reference, and parthenolide (**Pa**), a SL with proven antitumor activity and with previous clinical trials in humans. The results of molecular docking interactions between selected pharmacological receptors and incomptine A (**IA**) are shown in Figure 2 and Table 2. It was found that **IA** showed the best interaction with fructose-bisphosphate aldolase (ALDOA) and microsomal glutathione S-transferase 1 (MGST1) with binding energies of −8 kcal/mol and −8.4 kcal/mol, respectively. In the case of L-lactate dehydrogenase A (LDHA) and L-lactate dehydrogenase B (LDHB), a good binding energy was shown with −6.8 kcal/mol and −6.9 kcal/mol, respectively; followed by nuclear factor kappa-light-chain-enhancer of activated B cells (NFkB, −5.7 kcal/mol) and BCL-2 regulator protein (BCL2A1, −4.8 kcal/mol).

The best pose obtained in LDHA showed interaction with Asn137 and Thr247 in the binding site with **IA** (Figure 2A). With LDHB, incomptine A (**IA**) showed a similar pose inside the cavity interacting with Asn139 through two polar contacts, but lost a polar contact with the oxirane moiety (Figure 2B). Incomptine A (**IA**) showed a strong binding affinity with fructose-bisphosphate aldolase (ALDOA) by forming a strong interaction with Ser271 by donating three hydrogen bonds simultaneously to this residue and two hydrogens bonds with Arg42 and one extra polar contact between the epoxide group and Lys146 (Figure 2C). In BCL-2A1, the pose of **IA** had the lowest binding energy, however, it showed three conventional hydrogen bonds with Lys77 and two simultaneous interactions with Asn51 through the =O of lactone and =O of ester (Figure 2D). The docked complex of **IA**-NFkB showed two polar contacts between –O– from lactone and Gln199, and -O– from epoxide and Asp97 (Figure 2E). Finally, we observed the last selected target MGST1 and the docked pose displayed two polar contacts, the first polar interaction among =O from ester and Arg88 and the second important bond between –O– from oxirane and Arg130 (Figure 2F). Thereby, based on the in silico analysis, **IA** has greater interaction at two receptors (ALDOA and MSTG1) and suggest that it is a candidate for a multi-target cancer drug.

## 3. Discussion

In the course of our continuing search for potential antitumoral agents [15,16,17,18] from plants commonly used in traditional medicine, in this research, the aerial parts of *Decachaeta incompta* were extracted by percolation with dichloromethane. Then, the resultant extract was evaluated in the brine shrimp acute lethality test (BSLT) considering that this bioassay has been used as a bioindicator of the presence of potential antitumor agents in the crude extracts of plants [17,18,19]. The results obtained (Table 1) show that the **DEDi** induced lethality with a LD_50_ 90.2 µg/mL, was therefore lethal or active according to the classification system devised by Meyer [19]. The results obtained in the BSLT and antilymphoma activity showed that the **DEDi** is a source potential of antitumor agents [15,16,17,18,19]. Therefore, the dichloromethane extract was subjected to silica gel column chromatography (CC), and five fractions were collected. Based on TLC analysis, two fractions of these were combined, and after purified by CC to obtain incomptine A (Figure 1, **IA**). In the BSLT assay, incomptine A (**IA**) showed significant lethality (Table 1) with a LD_50_ 16.7 µg/mL, which was five times more active than the origin extract and nearly two times more active than methrotexate (**MTX**). The results indicate that **IA** may be a potential antitumor agent. Once the lethality of the **DEDi** and **IA** on *Artemia salina* larvae were observed, we focused on evaluating their activity as antitumor agents, specifically in a lymphoma model. Thus, the evaluation of the effects of **DEDi** and incomptine A (**IA**) on the axillary and inguinal lymph nodes of female Balb/c mice injected with U-937 (leukemic monocyte lymphoma) cell line showed significant activity with an EC_50_ value of 75 mg/kg and 2.4 mg/kg to **DEDi** and **IA**, respectively. The antilymphoma activity of dichloromethane extract of *D. incompta* was in agreement with the previously reported data for other medicinal plants with significant antitumor properties including *Schinus mole* (EC_50_ 99.4 mg/kg) and *Annona microprophyllata* (EC_50_ 285.69 mg/kg) [17,18]. The antilymphoma effect of incomptine A (**IA**) was 31-fold more active than the origin extract and comparable to methotrexate (**MTX**), an antilymphoma drug used as the positive control. These data suggest that **IA** may play an important role in the antitumor activity of the *D. incompta* extract and additionally confirms that SL may be used as a potential specialized metabolite for the development of novel antitumor drugs [2]. Since the **DEDi** and **IA** showed significant BS lethality as well as antilymphoma activity, the results suggest that the brine shrimp lethality test may be used as an economic, rapid, and reliable bioassay to obtain antilymphoma agents [17,18,19]. It is worth mentioning that SLs are a group of sesquiterpenoids commonly isolated from the flowers and leaves of Asteraceae species. In this sense, parthenolide (**Pa**) is a SL derived from *Tanacetum parthenium* aerial parts, which is structurally related with **IA** and has in vitro and in vivo antiproliferative activity on human cancer cells including SH-J1, MDA-MB-231, and COLO205 [8]. These properties of **Pa** are associated with the presence of the α-methylene-γ-lactone moiety, which exercises its effects by reacting with thiol groups available on the proteins and enzymes present in signaling pathways, especially the NF-kB pathway and STAT3, and through a change in the redox cell balance, among others [2]. Due the above, some clinical trials have been carried out using pathenolide (**Pa**) alone or combined for cancer treatment [8,20]. In addition, the results of the acute toxicity test were in agreement with the Globally Harmonized Classification System (GHS), and indicated that **DEDi** was category 4, while **IA** was category 3 [20]. The acute toxicity of incomptine A (**IA**) was close to methotrexate, an antilymphoma drug used currently in Mexico for the treatment of non-Hodgkin lymphoma. To our knowledge, this is the first report of the antilymphoma activity, brine shrimp lethality, and acute oral toxicity of incomptine A (**IA**) and the dichloromethane extract of the aerial parts of *D. incompta*.

In relation of molecular docking studies, parthenolide (**Pa**) was chosen to be recorded (Figure 3) considering that it is a SL with anticancer activity and has known effects on NFkB and MGST1 in in vitro and in vivo models [2,8,21,22]. Methotrexate (**MTX**) was recorded (Figure 4) to have the highest docking score into several pharmacological targets. **MTX** is widely known as an antilymphoma agent [10,16]. The molecular docking analysis with incomptine A (**IA**) versus parthenolide showed that **IA** had a high score value of affinity (Table 2) than that of parthenolide against LDHA, LDHB, and MGST1.

In the case of incomptine A (**IA**) versus methotrexate, both showed similar affinity (Table 2) against MGST1 and ALDOA.

The analysis (Figure 5) of the superimposed poses of incomptine A (**IA**), parthenolide (**Pa**), and methotrexate (**MTX**) versus six pharmacological targets showed a close interaction with the site receptor in all cases, varying in ΔG, affinity, and several amino acids (Table 2). This observation was in agreement with the antilymphoma activity of the three compounds. Finally, the molecular docking analysis suggests that antilymphoma properties of **IA** may be associated with the effects on the six pharmacological targets used including L-lactate dehydrogease A (LHDA), L-lactate dehydrogenase B (LDHB), fructose-bisphosphate aldolase (ALDOA), Bcl-2-regulator protein A1 (BCL-2A1), NF-kappa B p65 (RelA) homodimer (NFkB), and microsomal glutathione S-transferase 1 (MGST1). Additionally, these observations are in agreement with our recent in vitro results [15] and support additional evidence of the mechanism of action of incomptine A (**IA**).

## 4. Materials and Methods

### 4.1. Collection and Identification of Decachaeta incompta

*Decachaeta incompta* (D.C.) King and Robinson (Asteraceae) aerial parts were collected in Portillo Nejapa de Madero (16°36′00″ N, 95°59′00″ O) State of Oaxaca, Mexico. The plant was botanically authenticated by M Sc Abigail Aguilar and a specimen (voucher 15311) was deposited at the medicinal Herbarium IMSSM of the Instituto Mexicano del Seguro Social (IMSS).

### 4.2. Cancer Cell Line

U-937 cells (human leukemic monocyte lymphoma; ATCC: CRL 1593.2. Middlesex, London, UK) were provided by UIM en Genética Humana del Hospital de Pediatría de CMN S XXI, IMSS. Cell culture was tested for mycoplasma contamination using a MycoAlert mycoplasma detection kit (Lonza Pharma& Biotech, Walkersville, MD, USA).

### 4.3. Animals

Male and female BALB/c mice (25–30 g) were obtained from the animal house of the IMSS. These studies were conducted with the approval of the Bio-Ethical and National Scientifical Research Committees of the National Medical Center Siglo XXI from IMSS (Approval No: R-2018-785-111). Investigation using experimental animals was conducted in accordance with the official Mexican norm NOM 0062-ZOO-1999 [23] entitled technical specifications for the production, care, and use of laboratory animals. Animals were maintained with a 12 h light-dark cycle at 22 ± 2 °C at the controlled condition. They were fasted overnight, but tap water was available ad libitum until the start of the experiments.

### 4.4. Chemicals

Methotrexate, 3-(4,5-dimethylthiazol-2-yl)-2,5-diphenyltetrazolium bromide (MTT), dimethyl sulfoxide (DMSO), L-glutamine, penicillin/streptomycin, RMPI 1640 medium, acetonitrile HPLC grade, acetic acid HPLC grade were purchased from Sigma-Aldrich, USA. AR grade EtOH, dichloromethane, hexane, and MeOH were purchased from JT Baker, Mexico. Fetal bovine serum was purchased from Gibco, CdMX Mexico.

## 5. Isolation of Incomptine A of the Aerial Parts from *D. incompta*

### 5.1. Preparation of the Aerial Parts Extract

The air-dried aerial parts (25 g) were ground and extracted by percolation at room temperature with dichloromethane (350 mL). The extracts were concentrated under vacuum to yield 2 g of brown residue.

#### Dichloromethane Extract: Isolation and Purification

The dichloromethane extract (1.8 g) was subjected to column chromatography (CC) over silica gel (20 g, 70–230 mesh, Merck) using hexane, and a mixture of dichloromethane-MeOH (7:3–5:5) to give five fractions (Fr_1_–Fr_5_). Fractions 3 and 4 were combined and resolved by CC over silica gel (20 g) using a mixture of solvents, dichloromethane in MeOH (7:3–5:5) to yield incomptine A (95 mg). Incomptine A (**IA**) was identified by comparison (^1^H-NMR and TLC) with authentic samples disposed in our laboratory [3,4]. TLC (R_f_: 0.60; silica gel, CHCl_3_: EtOAc, 95:5, *v*/*v*). The ^1^H-NMR spectra (CDCl_3_): *d* = 6.40 (dd, *J* = 2.0, 0.5 Hz, H-13), 6.14 (d, *J* = 11.5 Hz, H-3), 5.80 (d, *J* = 2.0 Hz, H-13), 5.55 (dd, *J* = 11.5, 7.5 Hz, H-2), 5.31 (dq, *J* = 11.0, 1.5 Hz, H-5), 5.20 (ddd, *J* = 4.5, 3.0, 2.0 Hz, H-8), 5.01 (dd, *J* = 11.0, 1.5 Hz, H-6), 3.26 (dd, *J* = 7.5, 1.0 Hz, H-1), 2.94 (quin, *J* = 3.0, 1.5 Hz, H-7), 2.70 (dd, *J* = 14.5, 4.5 Hz, H-9), 2.02 (s, AcO), 1.88 (s, Me-15), 1.42 (s, Me-14), and 1.38 (ddd, *J* = 14.5, 3.0, 0.5 Hz, H-9).

### 5.2. Antilymphoma Test

Animals were randomly divided into eleven groups (six BALB/c mice per group) as follows: G1, G2, G3 [G3a, G3b, and G3c], G4 [G4a, G4b, and G4c], and G5 [G5a, G5b, and G5c]). For comparison, G1 was designated as the normal control group, which was neither inoculated with cancer cells nor treated with or **DEDi** or incomptine A or methotrexate. Lymphoma was induced according to Calzada et al. [17]. U-937 cells were injected intraperitoneally with 1 × 10^6^, U-937 cells/mouse to all mice in the G2, G3, G4, and G5 groups. Group G2 was reserved as the cancer control, so was not treated with any compounds but only with the vehicle (2% tween 80 in water). Twenty-four hours after, the animals in groups G3, G4, and G5 were treated with **DEDi**, incomptine A (**IA**), and methotrexate (**MTX**) at doses of 0.1, 1.0, and 10 mg/kg) orally, respectively. The treatment was continued for nine days. Animals were maintained under observation for 30 days, recording the daily survival. After completion of the experiment, all animals were sacrificed, and axillary and inguinal lymph nodes were removed and weighed. Antilymphoma activity of the samples was measured as the total weight of the lymph nodes and expressed in percent of inhibition. After the plot of percentage of inhibition against concentration was made, the best straight line was determined by regression analysis and the 50% effective inhibitory concentration (EC_50_) values were calculated. The regression equation and the coefficient of correlation were then derived from the curve.

### 5.3. Brine Shrimp Acute Lethality Test

Brine shrimp lethality tests were performed as described previously Meyer et al. [19]. The extracts and pure compounds were dissolved in 2 mL of ethanol. Then, 5, 50, and 500 μL of these solutions were transferred into 10 mL vials and the solvent was evaporated to obtain the final concentrations of the extract of 10, 100, and 1000 μg/mL (or 0.1, 1, and 10 μg/mL of pure compounds). After 10 *Artemia saline* larvae were introduced into each vial, the volume was adjusted with artificial seawater up to 5 mL. It should be noted that this procedure was performed in triplicate for each test. After 24 h, the *Artemia saline* larvae were counted. The lethal concentration 50 (LC_50_) values were obtained. LC_50_ of <100 μg/mL was considered lethal or active, moderately lethal when LC_50_ was between 100 μg/mL and 500 μg/mL, whereas LC_50_ value of >1000 μg/mL was considered as non-lethal or inactive.

### 5.4. Acute Oral Toxicity

The acute oral toxicity test was conducted in compliance with OECD guideline 423 for the testing of chemicals. Thirty-nine female BALB/c mice fasted overnight with free access to water ad libitum were randomly assigned into the following thirteen groups of three mice each. Control group received distilled water. Four groups received the **DEDi** at doses of 5, 50, 300, and 2000 mg/kg. In the case of pure compounds, the remaining eight groups received methotrexate (**MTX**) or incomptine A (**IA**) at the same doses of the extract. Subsequently, the mice were not fed for 4 h following administration. The signs of toxic effects and mortality were observed in the first 6 h after single treatment and then for 48 h mice were observed for neurotoxic signs such as dizziness, lethargy, or aggressiveness. Animal behavior was recorded during the subsequent 14 days, and the animals were finally sacrificed using a CO_2_ saturation chamber. A gross necropsy and an external and internal evaluation of the animals were completed. Furthermore, internal organs (stomach, intestine, liver, kidney, and spleen) were extracted and macroscopical observations were performed. Then, the median lethal dose (LD_50_) was calculated [20].

### 5.5. Statistical Analysis

The plot of percentage of inhibition or lethality or mortality against concentration was made; the best straight line was determined by regression analysis, and the EC_50_ and LD_50_ were calculated. All data were expressed as mean ± standard deviation of six measurements. Statistical analysis of data was performed using one-way ANOVA. A probability value of *p* < 0.05 and a correlation coefficient > 0.9700 were considered statistically significant. Differences between groups were analyzed by the Bonferroni and Dunnett post hoc test. Analyses were performed using GraphPad Prism Version 5.03 (GraphPad Software Inc., La Jolla, CA, USA).

## 6. Molecular Docking Analysis

### 6.1. Homology Modeling of LDH B Chain

Homology modeling was applied to build a 3D structure of the human L-lactate dehydrogenase B chain. The sequence was downloaded from the universal protein resource (Uniprot) with entry P07195 [24]. The L-lactate dehydrogenase A chain (PDB id: 4JNK) was used as a template and sequence of alignment. The sequence alignment and 3D-building were carried out with MODELLER 10.0 [25,26]. One hundred models were constructed and only one model was selected based on the best DOPE score. The final model was submitted to minimization under the MMFF94 forcefield to attempt to resolve the clashes and was used to carry out the molecular docking.

### 6.2. Molecular Docking of Incomptine A (**IA**), Parthenolide (**Pa**), and Methotrexate (**MTX**)

The incomptine A (**IA**) or parthenolide (**Pa**) or methotrexate (**MTX**) structure was created and prepared using MOE software [27]. Several three-dimensional structures involved in the treatment of cancers were retrieved from the Protein Data Bank (https://www.rcsb.org) (accessed on 21 October 2021) with the following accession codes: 4JNK (L-lactate dehydrogenase A chain, LDHA, 1.90 Å of resolution); 4ALD (Fructose-bisphosphate aldolase A, ALDOA, 2.8 Å of resolution); 5WHI (Bcl-2-related protein A1, BCL2A1, 1.69 Å of resolution); 2RAM (NF-kappa B p65, RelA, NFKB homodimer, 2.4 Å of resolution); and 5I9K (Microsomal glutathione S-transferase 1, MGST1, 3.5 Å of resolution). Molecular targets and incomptine A were submitted to MOE software. All water molecules and artifacts were removed from the crystallographic structures. Then, all polar hydrogen atoms were added, ionized in a basic environment (pH = 7.4), and Gasteiger charges were assigned.

The molecular docking experiments were carried out using AutoDock Vina with 20 modes and 16 exhaustiveness [28,29]. The active site of each target was covered with the proper size of the grid. The grid was centered at the following coordinates: LDHA (center_x = 43.652; center_y = 21.862, and center_z = 47.21) with grid dimensions of 16 × 16 × 16 points; LDHB (center_x = 43.652; center_y = 21.862, and center_z = 47.21) with grid dimensions of 16 × 16 × 16 points; ALDOA (center_x = 1.328; center_y = 29.617, and center_z = 42.887) with grid dimension of 14 × 14 × 14 points; *BCL2A1* (center_x = 10.161; center_y = 14.773, and center_z = 41.8) with grid dimensions of 18 × 18 × 18 points; *NFKB* (center_x = −0.746; center_y = 48.945, and center_z = 55.159) with grid dimensions of 22 × 22 × 20 points; and MGST1 (center_x = 40.639; center_y = 31.674, and center_z = 15.608) with grid dimensions of 14 × 14 × 14 points.

### 6.3. Docking Validation Protocol

The validation of the molecular docking results was conducted by re-docking the co-crystallized ligand in those receptors that present it (PDB ID: 4JNK, 4ALD, and 5I9K). The lowest energy pose of re-docking and the co-crystallized ligands were superimposed and it was observed whether it gained the same position, and its RMSD was calculated between these two superimposed ligands. The RMSD was within a reliable range of 2 Å.

## 7. Conclusions

In this study, we found that incomptine A (**IA**) possesses strong antitumor activity on leukemic monocyte lymphoma (U-937) in the mice model. Additionally, the in vitro, in vivo, and in silico results suggest that **IA** has pharmacological potential as an anticancer agent. In future studies, we plan to confirm its potential use as a therapeutic agent to combat non-Hodgkin lymphoma.

## Figures and Tables

**Figure 1 molecules-26-06646-f001:**
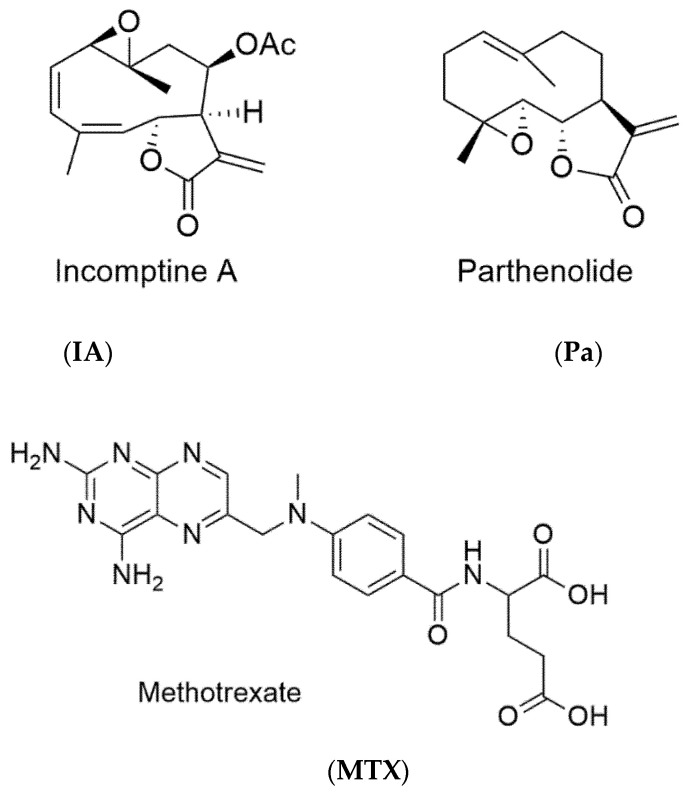
Structures of incomptine A (**IA**), parthenolide (**Pa**), and methotrexate (**MTX**).

**Figure 2 molecules-26-06646-f002:**
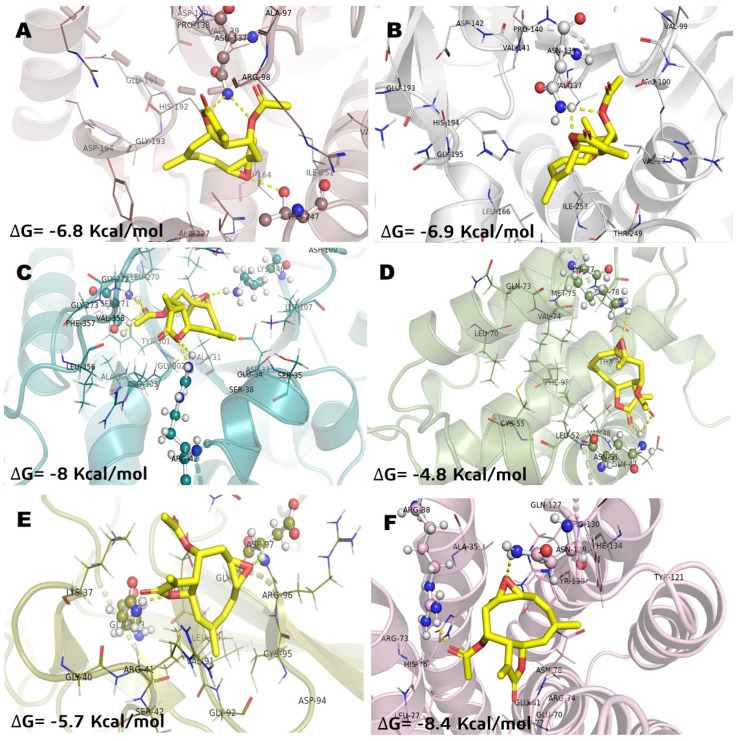
Molecular model of the virtual screening for incomptine A (**IA**) bound to (**A**) L-lactate dehydrogease A chain (LHDA); (**B**) L-lactate dehydrogenase B chain (LDHB); (**C**) Fructose-bisphosphate aldolase (ALDOA); (**D**) Bcl-2-regulator protein A1 (BCL-2A1), (**E**) NF-kappa B p65 (RelA) homodimer (NFkB); and (**F**) Microsomal glutathione S-transferase 1 (MGST1); ΔG = binding energy.

**Figure 3 molecules-26-06646-f003:**
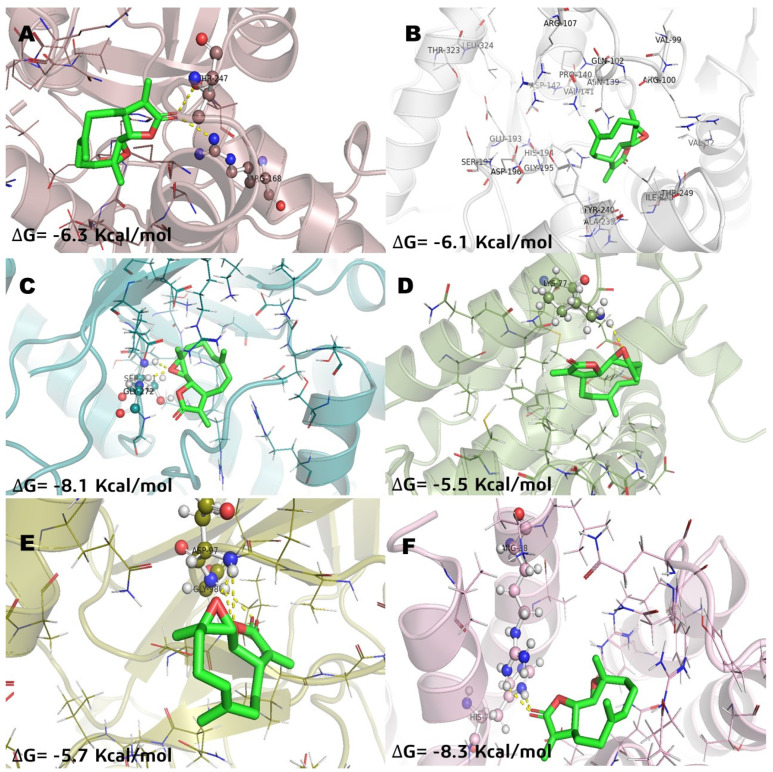
Molecular model of virtual screening for parthenolide (**Pa**) bound to (**A**) L-lactate dehydrogease A chain (LHDA), (**B**) L-lactate dehydrogenase B chain (LDHB), (**C**) Fructose-bisphosphate aldolase (ALDOA), (**D**) Bcl-2-regulator protein A1 (BCL-2A1), (**E**) NF-kappa B p65 (RelA) homodimer (NFkB), and (**F**) Microsomal glutathione S-transferase 1 (MGST1); ΔG = binding energy.

**Figure 4 molecules-26-06646-f004:**
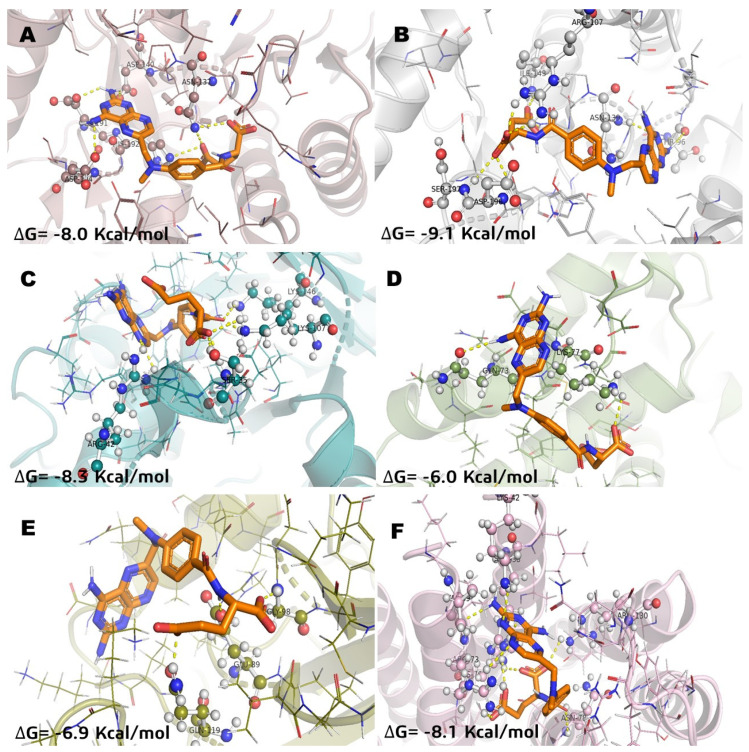
Molecular model of virtual screening for methotrexate (**MTX**) bound to (**A**) L-lactate dehydrogease A chain (LHDA), (**B**) L-lactate dehydrogenase B chain (LDHB), (**C**) Fructose-bisphosphate aldolase (ALDOA), (**D**) Bcl-2-regulator protein A1 (BCL-2A1), (**E**) NF-kappa B p65 (RelA) homodimer (NFkB), and (**F**) Microsomal glutathione S-transferase 1 (MGST1); ΔG = binding energy.

**Figure 5 molecules-26-06646-f005:**
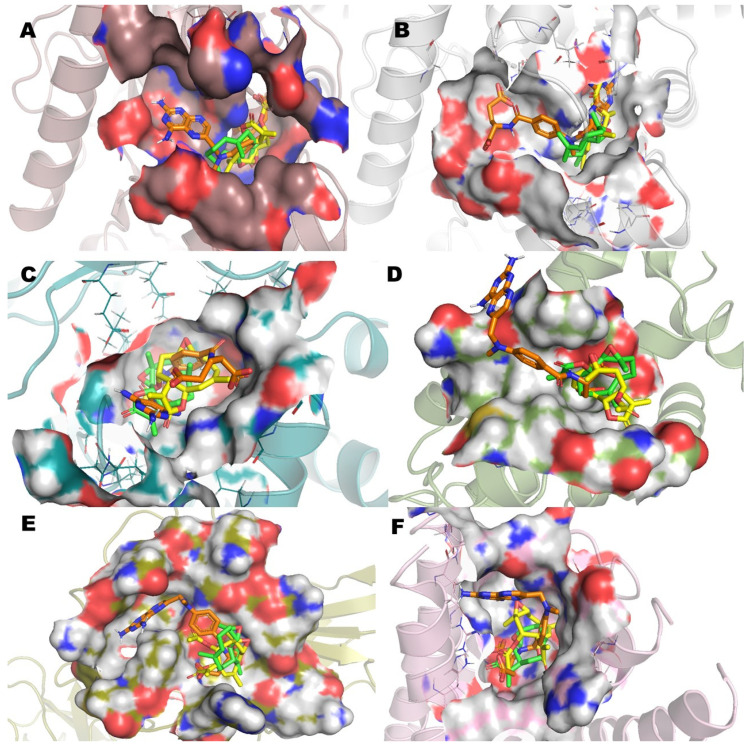
Superimposed poses of methotrexate (**MTX**), parthenolide (**Pa**), and incomptine A (**IA**) into the binding site bound to (**A**) L-lactate dehydrogease A chain (LHDA), (**B**) L-lactate dehydrogenase B chain (LDHB), (**C**) Fructose-bisphosphate aldolase (ALDOA), (**D**) Bcl-2-regulator protein A1 (BCL-2A1), (**E**) NF-kappa B p65 (RelA) homodimer (NFkB), and (**F**) Microsomal glutathione S-transferase 1 (MGST1).

**Table 1 molecules-26-06646-t001:** Antilymphoma, lethality, and toxic activities of incomptine A (**IA**) and the dichloromethane extract of the aerial parts from *D. incompta*.

Sample	Antilymphoma Activity EC_50_ (mg/kg) in Female Mice *	BS Lethality Test LD_50_ (µg/mL) *	Acute Oral Toxicity Test LD_50_ (mg/kg) in Female Mice *
**DEDi**	75.0 ± 1.2	90.2 ± 0.77	1134.0 ± 0.39
Incomptine A (**IA**)	2.4 ± 0.10	16.7 ± 0.85	149.0 ± 0.50
Methotrexate (**MTX**)	1.3 ± 0.26	24.6 ± 0.27	335.0 ± 0.39

* Correlation coefficient > 0.9700; Data are expressed as mean ± S.E.M (*n* = 6), *p* < 0.05; BS, brine shrimp; DEDi, dichloromethane extract of *Decachaeta incompta* aerial parts.

**Table 2 molecules-26-06646-t002:** ΔG (kcal/mol), Ki (mM), and receptor–ligand interactions from molecular docking.

Compound		Incomptine A (IA)	Parthenolide (Pa)	Methotrexate (MTX)
**LDHA**	ΔG	−6.8	−6.3	−8
	Ki	14.4	19.68	6.82
	aa residues	Asn137, Thr247	Arg 168, Thr247	Asn137, Asp140
				Glu191, Asp194, His192
**LDHB**	ΔG	−6.9	−6.1	−9.1
	Ki	13.54	22.3	3.43
	aa residues	Asn139	-	Asn139, Thr96, Ile143
				Asp142, Arg107, Ser197
**ALDOA**	ΔG	−8	−8.1	−8.3
	Ki	6.82	6.41	5.65
	aa residues	Ser271, Arg42	Ser271, Gly272	Lys107, Arg42
		Lys146		Lys146, Ser35
**NF-kB**	ΔG	−5.7	−5.7	−6.9
	Ki	28.62	28.62	13.54
	aa residues	Gln199, Asp97	Gly98, Asp97	Gln119, Gly98
				Glu89
**BCL-2A1**	ΔG	−4.8	−5.5	−6
	Ki	50.16	32.42	23.73
	aa residues	Lys77, Asn51	Lys77	Lys77, Gln73
**MGST1**	ΔG	−8.4	−8.3	−8.1
	Ki	5.31	5.62	6.41
	aa residues	Arg88, Arg130	Hist76, Arg38	Hist76, Arg130
				Hist76, Arg38, Lys42

Asp: aspartate; Asn: asparagine; Arg: arginine; Gln: glutamine; Lys: lysine; Thr: threonine; Ser: serine; Trp: tryptophan; Lys: lysine; His: histidine; Gly: glycine; Glu: glutamic acid; Ile: isoleucine; Val: valine. IA: incomptine A; Pa: parthenolide; MTX: methotrexate; LDHA: L-lactate dehydrogenase A; LDHB: L-lactate dehydrogenase B; ALDO: fructose biphosphate aldolase A; NF-kB: nuclear factor kappa-light-chain-enhancer of activated B cells; BCL-2A1: BCL-2 regulator protein; MGST1: microsomal glutathione S-transferase 1; ΔG: binding energy; Ki: theorical inhibition constants; aa: amino acid.

## Data Availability

The data presented or additional data in this study are available on request from the corresponding author.

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
