# Peer review of "Antilymphoma Effect of Incomptine A: In Vivo, In Silico, and Toxicological Studies"

_molecules, 2021, doi:10.3390/molecules26216646_

Round 1
Reviewer 1 Report
This is a for-runner of a study aimed at describing the anti-lymphoma activity of the sequiterpene lactone compound incomptine A (IA), isolated from the Mexican plant Decachaeta incompta. The authors have recently reported the in vitro activities, using different cell lines. Here they present in vivo and toxicology data and a molecular modeling analysis to try to evidence the molecular targets for the studied compound. The in vivo section is (potentially) interesting, whereas the modeling analysis is verbose and of a minor interest here (because it is completely speculative). Altogether, the study may be published but a major, very significant revision is needed.
Major comments:
- The first part of the introduction is not very good (up to ref (8)). Sesquiterpene lactones are extremely diversified, it is not very useful to compare IA with other randomly chosen sesquiterpene lactones in general. Why not centering the introduction on incomptine compounds, with a presentation of existing incomptines (A,B,C,D, heliangolide-type) and other natural products isolated from Decachaeta incompta.
- The organization of the manuscript is questionable. For example, paragraph 2.1 (up to line 115) should not be in the Results section. This is still the Introduction. The in silico analysis is extremely verbose and far too long. It must be reduced considerably (this is purely conjectural, without experimental data to validate the proposed targets).
- The in vivo anti-lymphoma activity is potentially interesting, but the data are poorly presented (limited to one EC50). The survival curves should be presented, to support the EC50 values cited in Table 1. This is the key point of the paper. It should be extensively described (rather than speculative modeling). At present the claim for “a strong anti-lymphoma activity of IA” (conclusion) is not established at all. Proper in vivo data should be presented.
- The molecular modeling analysis, performed randomly with several targets is highly questionable. Table 2 give deltaG values for the binding of methotrexate to LDHA/LDHB, although to my knowledge this reference compound (studied for decades) does not directly bind to these two proteins. Therefore, why IA with less favorable dG values could be considered as a potential ligand for these proteins? This part is excessively speculative and of little interest at present without experimental validation. Must be reduced.
Minor points
- Abstract: trough -> through (line 27)
- It would be convenient to split the introduction in two paragraphs (separate the intro after … [8]).
- Line 116: Briefly explain the basis of the brine shrimp acute lethality test. Why using this peculiar test and not a conventional lymphoma cell proliferation assay?
- Table 2: should mention the protein structures used for each target (PDB access codes).
- The molecular models for parthenolide (Fig 3) and methotrexate (Fig 4) MUST be removed. Totally useless. This is a paper on IA, not about the reference compounds.
- Line 260: lymphoma -> lymphoma (also line 398)
- Remove docking score from the abstract (useless). T
- he superimposed models in Figure 5 bring no useful information. They can be removed as well. This is just pretty images but without scientific value.
- Figure 6 is not essential (supplementary data)
Author Response
Comments and suggestion for Authors
Reviewer 1
Major comments:
Query 1.- The first part of the introduction is not very good (up to ref 8). Sesquiterpene lactones are extremely diversified, it is not very useful to compare IA with other randomly chosen sesquiterpene lactones in general. Why not centering the introduction on incomptine compounds, with a presentation of existing incomptines (A, B, C, D, heliangolide-type) and other natural products isolated from Decachaeta incompta.
Answer
In agree with the suggestion additional information was include about sesquiterpene lactones isolated from D. incompta including incomptine A analogs obtained by synthesis (lines 67 to 72 references 13 and 14).
The previous introduction (lines 41 to 66) doesn’t was deleted or changed considering that reviewer 2 is agree with this information. Delete this part may cause conflict with reviewer 2.
Query 2.- The organization of the manuscript is questionable. For example, paragraph 2.2 (up to line 115) should not be in the results section. This is still the introduction. The in silico analysis is extremely verbose and far too long. It must be reduced considerably (this is purely conjectural, without experimental data to validate the proposed targets.
Answer
Dear reviewer thanks for your comments. In this context lines 96-115 of result section were deleted.
In this case of the in-silico analysis doesn’t was changed considering:
1.- the format of molecules is free, 2.- reviewer 2 is agree with this information. 3.- Delete this part or change may cause conflict with reviewer 2. 4.- Previously by differential protein expression (reference 15) we demonstrated that LDHA, LDHB, ALDOANF-kB, BCL-2A1, and MGST1 are targets in U-937 cells. The information in silico result section provided additional support that incomptine A have as targets: LDHA, LDHB, ALDOA, NF-kB, BCL-2A1, and MGST1.
Query 3.- The in vivo anti-lymphoma activity is potentially interesting. But the data are poorly presented (limited to one EC50). The survival curves should be presented to support the EC50 values cited in table 1. This is the key point of the paper. It should be extensively described (rather than speculation modeling). At present the claim for “a strong anti-limphoma activity of IA” (conclusion) is not established at all. Proper in vivo data should be presented.
Answer.
Dear reviewer in this case we support our paper on basis of three aspects (in vivo, in silico and toxicological studies) using anti-lymphoma activity, BS lethality, acute toxicity, and docking study. The three first activates provided additional evidence that incomptine A have potential as antitumor agent and has an extensive support with docking search that was made using six targets recently defined by proteomic analysis. In a next contribution that is in process we will show iTRAQ of in vivo lymphoma model using different survival curves. Therefore, this information is reserved. For the other point since that reviewer 2 is agree with this form. Change this part may cause conflict with reviewer 2.
Finally, the claim that incomptine A have “ a strong antilymphoma activity” is support that result that its activity is close than methotrexate anti lymphoma drug used as positive control (lines 122-123 and 268-269 in results and discussion respectively).
Query 4. The molecular modeling analysis, performed randomly with several targets is highly questionable. Table 2 give delta G values for the binding of methotrexate to LDHA/LDHB, although to my knowledge this reference compound (studied for decades) does not directly bind to these two proteins. Therefore, why IA with less favorable dG values could be considered as a potential ligand for these proteins?. This part is excessively speculation and little interest at present without experimental validation. Must be reduced.
Answer.
Previously by differential protein expression (reference 15) we demonstrated that LDHA, LDHB are targets to incomptine A in U-937 cells. The cited result is agreement with in silico results and give additional support that LDHA and LDHB are affected by incomptine A therefore it could be considered as a potential ligand of these two proteins. In this sense this part was not changed.
Minor points.
1.-Abstract: trough à “through” (line 27)
2.- It would be convenient to split the introduction in two paragraphs (separate the intro after …..[8]).
3.-Line 116: Briefly explain the basis of the brine shrimp acute lethality test.
4.-Why using this peculiar test and not a conventionally lymphoma cell proliferation assay?
5.- Table 2: Should mention the protein structure used for each target (PDB access codes)
6.- The molecular models for parthenolide (Fig.3) and methotrexate (Fig 4) must be removed. Totally useless. This a paper on IA, not about the reference compound.
7.-Line 260: lymphoma-àlymphoma (also line 398)
8.-Remove docking score from the abstract (useless).
9.- The superimposed models in Figure 5 bring no useful information. They can be removed as well. This is just pretty images but without scientific value.
10.- Figure 6 is not essential (supplementary data).
Answer.
1.- trough was change as “through” (line 27)
2.- the introduction was separate after reference [8]
3.- The explain is in line 70 of introduction and 247 and 248 of discussion.
4.- Because “considering that this bioassay has been used as bioindicator of the presence of potential antitumor agents in crude extracts of plants and secondary metabolites”. In addition, cell proliferation assay was previously reported see reference [15].
5.- The PDB access codes are cited 7.2 section.
6 and 9.- Since reviewer 2 is in agree with the presence of this Figures these were conserved.
Delete this part may cause conflict with reviewer 2. Also, this Figures are illustrative of observed interactions.
7.- linphoma/limphoma were corrected
8.-docking score was removed
10.- Figure 6 was deleted.
Dr. Fernando Calzada
Reviewer 2 Report
Comments
1. In Table 2, explain why only these six receivers were chosen?
2. Explain in Table 2 what amino acids that are in bold mean.
3. In row 203, the correct statement is "IA has greater interaction (ΔG less than -8 Kcal/mol) at two receptors (ALDOA and MSTG1) and is a candidate for a multi-target cancer drug".
4. On line 547, change to "Molecular targets were submitted to MOE software".
5. Inform on line 551 the number of modes and exhaustiveness.
Author Response
Reviewer 2.
Query 1.- In Table 2. Explain why only these six receivers were chosen?
Answer.
In previous work (reference 15) by differential protein expression we found that LDHA, LDHB, ALDOA, NF-kB, BCL-2A1, and MGST1 were affected. Therefore, were chose for docking analysis.
Query 2.- Explain in Table 2 what amino acids are in bold mean.
Answer.
Was a mistake, these was corrected.
Query 3.- In row 203, the correct statement is “IA has greater interaction (G less than -8 kcal/mol) at two receptors (ALDOA and MSTG1) and is a candidate for a multi-target drug.
Answer.
The statement was corrected in agree with reviewer.
Query 4.- On line 547, change to “Molecular targets were submitted to MOE software.
Answer.
Line 547 was changed in agree with reviewer.
Query 5. Inform on line 551 the number of modes and exhaustiveness.
Answer.
This information was included.
Dr. Fernando Calzada
Round 2
Reviewer 1 Report
Most of my previous comments have not been addressed by the authors. The "in silico" part of the manuscript remains excessive, with speculative conclusions (in a journal dedicated to experimental works). I remain convenced that the computational aspects/data should be reduced, too speculative, not supported by experimental data.
I cannot agree with the publication of this Ms, in the present form.
Author Response
Comments and suggestion for Authors
Reviewer 1
The results were modified and conclusions were improved.
Major comments:
Query 1.- Most of my previous comments have not been addressed by the authors. The in silico part of the manuscript remains excessive, with speculative conclusion (in a journal dedicated to experimental works). I remain convinced that the computational aspects/data should be reduced too speculative, not supported by experimental data. I cannot agree with the publication of this Ms, in the present form.
Answer. Figures 3 to 5 were moved to supplementary material only Figure correspond with incomptine A (IA) interactions was conserved.
Also, introduction was shorted.
Dr. Fernando Calzada
